# Effects of Thermosonication on the Antioxidant Capacity and Physicochemical, Bioactive, Microbiological, and Sensory Qualities of Blackcurrant Juice

**DOI:** 10.3390/foods13050809

**Published:** 2024-03-06

**Authors:** Xiaokun Qiu, Jiajia Su, Jiangli Nie, Zhuo Zhang, Junhan Ren, Shiyi Wang, Yi Pei, Xihong Li

**Affiliations:** 1College of Horticulture and Landscape Architecture, Tianjin Agricultural University, Tianjin 300384, China; qxk333333@163.com (X.Q.); jjiasu@126.com (J.S.); njlnie@126.com (J.N.); zhuozhang1999@163.com (Z.Z.); 17848317782@163.com (J.R.); 13102034791@163.com (S.W.); 2State Key Laboratory of Food Nutrition and Safety, College of Food Science and Engineering, Tianjin University of Science and Technology, Tianjin 300457, China

**Keywords:** blackcurrant juice, thermosonication, physicochemical properties, bioactive compounds, antioxidant activity, microbiological safety

## Abstract

This study investigated the effects of thermosonication (TS) on the quality of blackcurrant juice, along with its physicochemical properties, bioactive compounds, antioxidant capacity, and microbiological and sensory qualities. The treatments included raw juice (RJ), pasteurized juice (90 °C, 1 min, PJ), and thermosonicated juice (480 W, 40 kHz at 40, 50, or 60 °C, for 10, 20, 30, or 40 min, TJ). The results indicated that the effects of pasteurization and thermosonication on the pH, total soluble solids, and titratable acidity of the juice were not significant (*p* > 0.05). However, the cloudiness, browning index, and viscosity were significantly increased (*p* < 0.05), and the color properties of the blackcurrant juice were improved. The total phenolic, flavonoid, and anthocyanin contents of TJ (treated at 50 °C for 30 min) were increased by 12.6%, 20.9%, and 40.4%, respectively, and there was a notable decline in ascorbic acid content after the pasteurization treatment, while the loss was minor in all TJ samples compared with RJ. The scavenging ability of 1,1-diphenyl-2-pyridyl and hydroxyl radicals increased to 52.77% and 50.52%, respectively, which were significantly (*p* < 0.05) higher than those in the RJ and PJ samples. In addition, both pasteurization and thermosonication resulted in a significant (*p* < 0.05) reduction in microbial counts, while there were no significant (*p* > 0.05) differences in the sensory parameters compared with the RJ samples. In conclusion, this study suggests that TS is an effective method that can be used as an alternative to pasteurization to improve the quality of blackcurrant juice.

## 1. Introduction

The blackcurrant (*Ribes nigrum* L.) is a berry that is extensively cultivated in Europe and also planted in Asia on a small scale [1]. The high levels of anthocyanins and phenolic compounds in blackcurrants contribute to their superior antioxidant properties, which offer several health benefits, including anti-inflammatory, antioxidant, and anticancer effects, along with disease prevention [2]. However, blackcurrants are not available to consumers as fresh fruit because they are highly perishable. In addition, consumption of fresh blackcurrants is limited due to their high acidity [3]. Thus, ripe blackcurrant fruit is commonly processed into juices, concentrates, and jams for year-round consumption instead of being eaten fresh [4,5]. In the case of fruit juice, the growth of microorganisms and enzyme activity can directly compromise quality. Therefore, appropriate treatments are necessary to ensure the microbiological safety and nutritional quality of fruit juice [6].

Conventional pasteurization has been the primary method for extending the shelf life of food products by inactivating microorganisms and enzymes, but the high temperatures of pasteurization treatment can adversely affect the sensory and nutritional qualities of fruit juices, such as color, phenolic contents, and vitamin levels [7,8]. Studies have reported that the pasteurization treatment reduces the vitamin C content in mango and watermelon juices and adversely impacts the color of pomegranate juice [9,10]. It also affects the aroma and nutritional value of citrus juices and leads to the degradation of phenolic compounds in blackcurrant juices, thereby diminishing the qualities of the juices [7,11]. The high temperatures during pasteurization can affect the bioactive compounds in fruit juices, whereas ultrasonic treatment has been shown to enhance the contents of bioactive compounds and antioxidant activities in fruit juices [12]. The generation and transmission of ultrasonic waves create fluctuations in fluid pressure and bubble formation, leading to cavitation (the aggregation and collapse of bubbles). The collapse of these bubbles generates mechanical effects, such as shear forces, shock waves, and micro-jets, which can induce various physical and chemical reactions. These reactions lead to the rupture of cells in the medium, along with the inactivation of bacteria and enzymes, and can improve or affect the physicochemical and nutritional qualities of the food materials [8,13].

TS treatment is an alternative processing method to traditional pasteurization for fruit juices, which refers to the combination of ultrasound and heat treatment, with both thermal and cavitation effects. It inactivates microorganisms and enzymes at milder temperatures and in shorter processing times, preserving nutrient stability and extending shelf life. The advantages of TS include reduced processing time, lower cost, environmental benefits, and energy savings [12,14,15]. It has been applied to various fruit juices, showing positive effects on improving the physicochemical properties, retaining bioactive compounds, and enhancing free radical scavenging. Studies have shown that TS improves the color properties of strawberry juice while maximizing its overall qualities and achieving bactericidal and enzyme inactivation [16]. TS treatment has increased the ascorbic acid content in purple cactus pear juice [17], the carotenoids and phenolic compounds in grape juice [18], the turbidity of black mulberry juice [19], and the viscosity of tomato juice [20]. It has also effectively inactivated PPO, POD, and PME in apple juice [21] and yeast in orange juice [22]. Furthermore, TS treatment has been shown to improve the qualities of cashew apple juice during storage and increase the antioxidant activities in amora (*Spondius pinata*) juice [14,23]. Some studies indicate a significant increase (*p* < 0.05) in TPC levels and antioxidant capacity in blackcurrant juice after ultrasound treatment [24]. However, the impact of TS treatment on the qualities of blackcurrant juice has rarely been reported. Therefore, this study aims to investigate the effects of TS treatment on the physicochemical properties, bioactive compounds, antioxidant capacity, microbial count, and sensory parameters of blackcurrant juice.

## 2. Materials and Methods

### 2.1. Blackcurrant Juice Preparation and Treatments

Completely mature and high-grade blackcurrants were harvested from Jixi (45°17′42″ N, 130°58′08″ E, altitude 230 m), Heilongjiang Province, China. After cleaning with distilled water, the blackcurrant fruit and drinking water were blended at a ratio of 1:1 and transferred to an electric juicer (MJ-PB12 Easy 219, Midea Group Co., Ltd. China) for approximately 30 s to produce blackcurrant pulp. Then, the fruit juice was filtered through four-layered sterile gauze (Jianerkang, Medical Technology Co., Changzhou, Jiangsu, China), and then it was divided into three parts for different treatments: untreated or raw juice (RJ), pasteurized juice (PJ), and thermosonicated juice (TJ).

Pasteurization treatment: Blackcurrant juice (200 mL) was placed in a 90 °C water bath (Li-Chen Bang Xi Instrument Technology Inc., Shanghai, China) for sterilization for 1 min, then immediately cooled in an ice-water bath and stored at 4 °C for further analyses.

Thermosonication treatment: Blackcurrant juice (200 mL) was treated in an ultrasonic cleaner (SB-4200 DTD, Xinzhi Biotechnology Inc., Ningbo, China) under conditions of 40 °C, 50 °C, and 60 °C, with a frequency of 40 kHz and power of 480 W, for durations of 10, 20, 30 and 40 min, designated as TS 40-10, TS 40-20, TS 40-30, TS 40-40, TS 50-10, TS 50-20, TS 50-30, TS 50-40, TS 60-10, TS 60-20, TS 60-30, and TS 60-40, respectively. The treated juices were then cooled and stored at 4 °C for subsequent use. The processes and treatments are illustrated in Figure 1 and listed in Table 1.

### 2.2. Physicochemical Analysis

#### 2.2.1. pH, Total Soluble Solids (TSS), and Titratable Acidity (TA)

The pH, TSS, and TA were determined following the methods used in [9], with slight modifications. The pH value of the blackcurrant juice samples was determined by a digital pH meter (PH−828, Sigma Instruments Group Co., Ltd., Dongguan, China). The TSS was determined using a handheld digital refractometer (AK002B, Meiyou Technology Co., Ltd., Shenzhen, China), and the results were expressed as °Brix. Titration was used to assess the titratable acidity; the blackcurrant juice samples (1 mL) were placed in a 100 mL volumetric flask with distilled water to fix the volume, shaken well, and let stand for 30 min, before being titrated with the calibrated NaOH solution (0.1 mol/L) and calculated as percentage citric acid.

#### 2.2.2. Cloudiness and Browning Index (BI)

The cloudiness and BI of the blackcurrant juice were determined by the methods described in [25]. The blackcurrant juice (5 mL) was centrifuged (TGL-16, Shu Ke Technology Co., Ltd., Chengdu, Sichuan, China) at 5000× *g* for 10 min at 25 °C, and the absorbance of the centrifugal supernatant was measured using a microplate reader (Spectra MAX 190, MEGU Molecular Instruments Co., Ltd., Shanghai, China) at 660 nm, with deionized water as a control. For the browning index, the obtained supernatant (5 mL) was mixed with 5 mL of ethanol and centrifuged again under the same conditions. The absorbance was measured at 420 nm.

#### 2.2.3. Viscosity

The viscosity of the juices was determined following the method described in [26], with slight modifications. It was determined by rotor No. 2 of a digital display viscometer (Li-Chen Bang Xi Instrument Technology Inc., Shanghai, China). Blackcurrant juices (100 mL) were placed in a beaker at a temperature of 25 °C ± 1 °C, and the viscosity was determined at 30 rpm, using distilled water as a control, and the results were expressed as centipoise (cP).

#### 2.2.4. Color Parameters

The juices (10 mL) were measured by a colorimeter (WR-18, Weifu Optoelectronic Technology Co., Shenzhen, China) using a light source (D65) and an observation point (10 °). The values of L*, a*, and b* were recorded to indicate color according to the method described in [27], with slight modifications. The total color difference (ΔE) was calculated to evaluate color changes after the treatment using Equation (1), where the parameters L_0,_ a_0_, and b_0_ are the initial values for the untreated samples.
(1)$$\Delta E={\left[{\left(L^{*}-L_{0}\right)}^{2}+{\left(a^{*}-a_{0}\right)}^{2}+{\left(b^{*}-b_{0}\right)}^{2}\right]}^{1/2}$$

### 2.3. Bioactive Compounds

#### 2.3.1. Total Phenolic Content (TPC)

The TPC of the juice samples was determined using the method described in [8], with slight modifications. Juice samples (0.5 mL), Folin–Ciocâlteu reagent (1 mL, 10%), and sodium carbonate (2 mL, 20%) were added and mixed well after adjusting the volume to 10 mL with distilled water. Then, the mixture was left at 30 °C ± 1 °C for 60 min under dark conditions. The absorbance of the juice samples was measured at 760 nm, and the results were expressed as mg of gallic acid equivalents (GAE) per 1 mL of sample.

#### 2.3.2. Total Flavonoid Content (TFC)

The TFC was measured using the AlCl_3_ method [28], with slight modifications. Briefly, the juice samples (1 mL) were mixed with sodium nitrite solution (0.3 mL, 5%) in test tubes and allowed to react for 5 min. Then, the aluminum chloride solution (0.6 mL, 10%) was added, and the reaction continued for another 5 min. Subsequently, the sodium hydroxide (1 mL, 1 M) was added, and the mixture solution was brought up to a final volume of 10 mL with distilled water. The absorbance was measured at 510 nm, with the results expressed as mg of catechin equivalents (CAE) per 1 mL of juice samples.

#### 2.3.3. Total Anthocyanin Content (TAC)

The TAC was determined by the pH differential method [29], with slight modifications. The blackcurrant juice samples (1 mL) were placed in two volumetric flasks (10 mL). Then, at pH 1.0 [(0.2 mol/L KCl):(0.2 mol/L HCl) =25:67, volume ratio] and pH 4.5 [(0.2 mol/L NaAc·3H_2_O):(0.2 mol/L HAc) = 1:1, volume ratio], the buffer was set to 10 mL and mixed evenly. The mixed solution (200 μL) was taken, and the absorbance was measured at 510 and 700 nm. The blank control group was 1 mL of deionized water and 9 mL of corresponding buffer solution. Equations (2) and (3) were used for calculating the anthocyanin content, as follows:(2)A=[(A510−A700)pH1.0−(A510−A700)pH4.5]
(3)ACY=A⋅V⋅n⋅449/m⋅29,600
where V represents the total volume of juice (mL), n is the dilution ratio, 449 refers to the relative molecular weight of cyanidin-3-O-glucoside (C3G), m represents the mass of raw blackcurrant fruit (g), and 29,600 is the extinction coefficient of C3G. The sample content was expressed as mg/100 g (C3G).

#### 2.3.4. Ascorbic Acid Content (AAC)

The AAC of the blackcurrant juices was determined by visual titration with 2,6-dichlorophenol indophenol dye, using the method described in [30]. The juice samples (10 mL) were added to a 100 mL volumetric flask, and the volume was made up with an oxalic acid solution (0.2%), before being shaken well and filtered for use. The filtrates (10 mL) were titrated with a standardized dye solution of 2,6-dichlorophenol indophenol to a pale pink endpoint for at least 15 s. The results were calculated using Equation (4) and expressed as mg/100 mL of each sample.
(4)$$AAC=V\cdot (V_{1}-V_{0})\cdot \rho \cdot 100/Vs\cdot m$$
where V_1_ is the volume of dye consumed for titration of one sample (mL), V_0_ is the volume of dye consumed for the blank titration (mL), ρ is the mass of ascorbic acid equivalent to 1 mL of dye solution (mg/mL), Vs is the volume of sample solution taken during the titration (mL), V is the total volume of the sample extract (mL), and m is the sample mass (g).

### 2.4. Antioxidant Capacity

#### 2.4.1. DPPH Radical Scavenging Ability

The DPPH radical scavenging ability was determined by referring to the method described in [31], with minor modifications. The juice samples (2 mL) were mixed with 2 mL of DPPH solution (0.2 mmol/L ethanol solution), and the absorbance at a wavelength of 517 nm was measured after reaction at 25 °C ± 1 °C for 30 min. The control group performed the same operation with distilled water instead of juice samples. The DPPH free radical scavenging capacity was calculated as follows:(5)$$DPPH\left(\%\right)=(A_{1}-A_{0})/A_{1}$$
where A_1_ and A_0_ are the assay and control absorbance values, respectively.

#### 2.4.2. Hydroxyl Radical (·OH) Scavenging Ability

The (·OH) scavenging ability of the juices was determined by the method described in [32], with slight modifications. The juice samples (2 mL) were mixed with ferrous sulfate solution (2 mol/L, 2 mL) and salicylic acid–ethanol solution (2 mol/L, 2 mL). Subsequently, hydrogen peroxide solution (1 mmol/L, 2 mL) was added to the mixture, which was placed in a 37 °C ± 1 °C water bath for 30 min. The absorbance was measured at 510 nm, with deionized water as a control.
(6)$$\cdot OH(\%)=[1-(A_{1}-A_{2})/A_{0}]$$where A_0_, A_1_, and A_2_ are the absorbance of the control, assay, and blank samples, respectively.

### 2.5. Microbiological Analysis

The microbial levels in the juice samples were evaluated to ensure the safety of the blackcurrant juice. *E. coli*, *Salmonella*, and *Staphylococcus aureus* not were detected in any of the fruit juices. The total bacterial count and the yeast and mold were counted using the spread plate method [33]. The juice samples were serially diluted to 10^−1^–10^−3^ with normal saline; then, the treated juice samples were inoculated on agar plates and incubated for 48 h at 37 °C ± 1 °C, and the total bacterial counts were recorded. Meanwhile, the yeast and mold were counted using potato dextrose agar and incubated for 120 h at 25 ± 1 °C. Then, the growth plates were examined to count the number of colonies, and the results were expressed as log CFU/mL.

### 2.6. Sensory Evaluation

The assessment of sensory qualities is critical to consumer acceptance of the juices. The method described in [2] was used to evaluate the qualities of the juice. Our evaluation team consisted of 20 trained food science professionals, including 10 men and 10 women. The panelists were instructed to rinse their mouths with water between sample evaluations. The sensory qualities of the blackcurrant juice samples were assessed, including color, taste, flavor, mouthfeel, and overall acceptability. The results were scored using a 9-share joy scale (9 = like extremely to 1 = dislike extremely).

### 2.7. Statistical Analyses

All experimental treatments were repeated three times, and the average results were obtained and expressed as the mean ± standard deviation. The data were analyzed by one-way ANOVA using SPSS 25 software. Differences between treatments were determined using Duncan’s multiple pole test, with the significance level set at *p* < 0.05. Origin 2018 was used to plot the diagrams. The images were analyzed using MATLAB 2016b to retrieve the red, green, and blue (RGB) colors.

## 3. Results and Discussion

### 3.1. Physicochemical Analysis

#### 3.1.1. pH, TSS, and TA

The effects of all treatments on the pH, TSS, and TA of blackcurrant juices are shown in Figure 2a–c, respectively. The pH values of RJ and PJ were 2.86 ± 0.07 and 2.91 ± 0.17, respectively, while the pH range of TJ was 2.84 ± 0.11 to 2.88 ± 0.09, which was not statistically significant (*p* > 0.05) when compared with RJ and PJ. This suggests that pasteurization and TS have no significant (*p* > 0.05) effect on the pH values of blackcurrant juices. Minor variations in the pH of the TJ samples could be attributed to chemical reactions arising from new chemical components produced in the juice [34]. The TSS of the juices was not significantly different (*p* > 0.05) from RJ (12.90 ± 0.30) after pasteurization (12.87 ± 0.21) and thermosonication (12.87 ± 0.25~13.00 ± 0.62). TSS is a reflection of soluble sugars, and the slight change in the TSS of fruit juices after TS may relate to the sonic dissolution of some fructose during cavitation [35]. The same result was found in TA, where PJ (0.35 ± 0.04) with TJ (0.33 ± 0.03~0.35 ± 0.04) had no significant effect (*p* > 0.05) on the TA of the juice compared with RJ (0.33 ± 0.04). This may be because the ultrasound energy levels applied to the juice samples did not alter the macromolecular structure associated with these physicochemical properties at the microscopic level. Our results are supported by studies on the thermosonication and pasteurization of porcine plum juices [36].

#### 3.1.2. Cloudiness and Browning Index (BI)

As shown in Figure 2d, the cloudiness of PJ (21.2 ± 0.32) significantly (*p* < 0.05) decreased compared with RJ (32.2 ± 0.61), whereas the cloudiness of juice samples treated with TJ (33.5 ± 1.03~37.8 ± 0.55) significantly (*p* < 0.05) increased. This could be attributed to the breakdown of large substances such as pectin, cellulose, hemicellulose, proteins, lipids, and other suspended particles present in the juice into smaller substances by cavitation of the TS bubbles, increasing the juice’s cloudiness [6]. Our findings were consistent with the results of thermosonication on grapefruit juices and black mulberries [18,19]. As shown in Figure 2e, the trend of the cloudiness was slightly different, with a significant (*p* < 0.05) increase in the browning index in both PJ (0.61 ± 0.04) and TJ (0.46 ± 0.05~0.54 ± 0.02) samples compared with RJ (0.38 ± 0.04). In our study, the BI increased slightly with the prolongation of TS and the increase in temperature, but this increase was not statistically significant (*p* > 0.05). The increase in the BI of juices due to thermosonication and pasteurization may be attributable to the occurrence of the Maillard reaction or the decomposition of ascorbic acid [16,23]. This finding is consistent with previous studies on the BI of mild-temperature ultrasound on orange juice and the effects of heat treatment and ultrasound on mango juice [37,38].

#### 3.1.3. Viscosity

The results of the viscosity determination for juices subjected to different treatments are shown in Figure 2f. The RJ samples were 7.12 ± 0.05 cP, which increased to 8.41 ± 0.06 cP for PJ, representing a significant (*p* < 0.05) difference compared with RJ. The TS treatment similarly elevated the viscosity of the juices to 7.92 ± 0.11~8.17 ± 0.16 cP, and this difference was significant (*p* < 0.05). Nevertheless, there were no significant (*p* > 0.05) differences in the viscosity of the blackcurrant juice samples with the increase in TS temperature and time. This increase could be due to the enhanced solubility of pectin in the cell wall because of cavitation during TS treatment [34]. The outcomes of our study are supported by the findings of a previous study on the TS of tomato juice [20].

#### 3.1.4. Color Parameters

The color of the juice is one of the most prominent quality parameters for consumer acceptance. L* indicates lightness; a* is the red–green axis, a positive value of a* means red, and the opposite means green; b* is the yellow–blue axis, +b* is yellow, and −b* is blue. The values of the color parameters (L*, a*, and b*) are shown in Table 2. Compared with RJ (23.81 ± 0.23, 1.58 ± 0.07, −2.46 ± 0.12), the L* values increased significantly (*p* < 0.05), the a* values decreased significantly (*p* < 0.05), and the b* values were not significantly (*p* > 0.05) different in PJ (25.83 ± 0.29, 1.31 ± 0.04, −2.11 ± 0.09) and TJ (25.06 ± 0.22~25.79 ± 0.33, 0.82 ± 0.04~1.45 ± 0.07, −2.45 ± 0.11~−2.21 ± 0.09). Variations in TS temperature and time did not significantly (*p* > 0.05) affect the L* values of the juice samples. However, the b* values of the juice samples decreased gradually with changes in TS temperature and time, and the temperature had a greater effect than time. Similar correlation results were found for the L* and b* values in strawberry juice [16]. The increase in color parameters concerns ultrasonic cavitation, which causes partial precipitation of suspended and unstable particles in the juice, as well as decomposition of pigments, resulting in the formation of other colored substances [36]. In addition, there were differences in ΔE values between PJ (2.07 ± 0.06) and TJ (1.26 ± 0.23~2.12 ± 0.17) samples, which gradually decreased with increasing TS temperature, but this difference did not result in a change in juice color that was visible to the naked eye. The reason for the increase in the ΔE values of the juices may be attributed to temperature leading to the Maillard reaction [19]. Furthermore, we include the RGB and HEX attributes in Table 2 to better identify the effects of different treatments on juice color.

### 3.2. Bioactive Compounds

The antioxidant capacity is correlated with the contents and activities of bioactive compounds in fruits and vegetables. In this study, the bioactive compounds (TPC, TFC, TAC, and AAC) and free radical scavenging capacity of blackcurrant juice were evaluated [39].

#### 3.2.1. Total Phenolic Content

Phenolic compounds are potent antioxidants that prevent oxidation of cells and play a crucial role in human health [40]. Furthermore, phenolic compounds are also important in the color changes of fruits and vegetables [41]. The results of the TPC are presented in Figure 3a, where TPC was 5.25 ± 0.05 and 4.94 ± 0.04 GAE mg/mL in RJ and PJ samples, respectively, with a significant (*p* < 0.05) increase observed in the TPC of TJ samples (5.36 ± 0.11~5.94 ± 0.09). The experiment demonstrated that TS significantly (*p* < 0.05) elevated the TPC in the juice, with higher ultrasound times resulting in higher TPC levels in the blackcurrant juice samples. While there was an increase in TPC with increasing temperature, it was not statistically significant (*p* > 0.05) within the range of 50 to 60 °C, suggesting that maintaining an appropriate temperature during TS is essential for preserving the TPC levels in blackcurrant juice. The higher TPC recorded in the TJ samples than in the RJ and PJ samples may be attributed to the increase in ultrasound time, which creates a shear force that causes a continuous disruption of the cell wall, effectively increasing the rate of diffusion of phenolic compounds, leading to an increase in their release [42,43]. In addition, TS presented a higher TFC, probably because phenolics are present in soluble form in plant vesicles or are bound to pectin, cellulose, hemicellulose, etc., in the cell wall; the cavitation generated by ultrasound through the periphery of the colloidal particles enhances the release of these compounds from the cell wall into the juice solution [17]. The outcomes of our study are supported by reports on the TS of purple cactus pear juice, blackberry juice, and red grape juice [17,25,44].

#### 3.2.2. Total Flavonoid Content

Flavonoids are important natural bioactive compounds with antioxidant properties [45]. As shown in Figure 3b, the trend of TFC was similar to that of TPC in blackcurrant juice. TFC was 2.12 ± 0.03 and 2.03 ± 0.04 RE mg/mL for the RJ and PJ samples, respectively, and it was significantly (*p* < 0.05) higher in TJ samples (2.23 ± 0.07~2.56 ± 0.04). The highest TFC was at 2.56 ± 0.04 mg/mL in the TS 50-30 sample, which was 1.26 times higher than that of the PJ sample and 1.21 times higher than that of the RJ sample. In all TS treatments, the TFC in the juice samples increased with increasing temperature and time; however, the effect of temperature was less than that of time. In the range of 50 to 60 °C, the increase in temperature did not have a significant (*p* > 0.05) effect on TFC, indicating that TS (50 °C, 30 min) had a positive effect on TFC in blackcurrant juice. Pasteurization has been shown to reduce TFC in pineapple juice [46], while ultrasound combined with pasteurization treatment had a better effect on TFC in the test samples than pasteurization alone [47]. The increase in TFC was related to the production of free radicals during ultrasound, or due to the rupture of the cell wall leading to the hydroxylation of phenolic compounds in the aromatic ring undergoing a hydroxylation reaction, which transforms the bound configuration of the phenolic compounds to the free form [48]. In this study, there was a significant (*p* < 0.05) difference between the effects of thermosonication and pasteurization treatments on the juices’ TFC. In other words, all TS treatments were superior to the pasteurization treatment in terms of TFC. This suggests that TS is more effective than pasteurization in enhancing the TFC of blackcurrant juice, and our results are similar to the changes in TFC reported in studies on the TS of elephant apple (*Dillenia indica*) juice [48].

#### 3.2.3. Total Anthocyanin Content

Anthocyanin is a water-soluble pigment that is responsible for the color of plant tissues and is also a highly abundant bioactive compound with antioxidant properties in blackcurrant fruit [49,50]. As shown in Figure 3c, the TAC of RJ and PJ was 1.13 ± 0.03 and 1.01 ± 0.06 Cyc mg/mL, respectively; however, the TJ (1.25 ± 0.04~1.59 ± 0.02) had significantly (*p* < 0.05) increased TAC compared to the PJ samples, with the highest anthocyanin content (1.59 ± 0.02 Cyc mg/mL) in the TS 50-30 samples. The TAC in the blackcurrant juice samples increased and then decreased with temperature and time, but time had a greater effect than temperature. The increase in TAC in fruit juices may be due to an increase in phenolic contents. In addition, cavitation generated during ultrasonic inactivates some oxidation-related enzymes (e.g., polyphenol oxidase, responsible for enzymatic browning), which also contributes to the increase in TAC in fruit juice [51]. For the changes in TAC content, our findings are consistent with the effects of high-intensity TS treatment on spinach juice [52]. However, negative effects of ultrasound have been found in other studies. For instance, the anthocyanin content decreased after ultrasonic treatment in strawberry juice [16]. The reason provided was the degradation of anthocyanins induced by the formation of hydrogen peroxide during the oxidation of ascorbic acid, because of the instability and easy breakdown of ascorbic acid at high temperatures.

#### 3.2.4. Ascorbic Acid Content

Figure 3d shows the effects of different treatments on AAC in blackcurrant juice. Compared with RJ (3.78 ± 0.24), PJ (2.37 ± 0.11) and TJ (2.97 ± 0.14~3.56 ± 0.08) contained significantly (*p* < 0.05) less ascorbic acid, and the AAC of PJ was much lower than that of TJ. The AAC of the juice samples gradually decreased with increasing temperature, and the reason for this is that ascorbic acid is unstable or decomposes easily at high temperatures. On the other hand, the gradual decrease in AAC with increasing TS time may relate to the extreme physical conditions generated by the cavitation collapse of bubbles during sonication. Furthermore, the degradation of ascorbic acid may relate to the oxidation reaction caused by the interaction of free radicals generated during ultrasound [53]. In conclusion, compared with pasteurization, TS treatment was more effective in maintaining the ascorbic acid levels in blackcurrant juice, similar to the results observed in the thermosonication of watermelon juice [54].

In summary, TS either enhanced or maintained higher levels of bioactive compounds in blackcurrant juice compared with pasteurization. These included TPC, TFC, TAC, and AAC, which are classes of bioactive compounds with positive antioxidant effects on health. This suggests that TS has a more positive effect on the retention of bioactive compounds in blackcurrant juice than pasteurization.

### 3.3. Antioxidant Capacity

#### 3.3.1. DPPH Radical Scavenging Ability

The scavenging capacity of the different treated blackcurrant juice samples for DPPH free radicals is shown in Figure 3e. The scavenging capacity of the PJ samples for DPPH was 38.27%, which was significantly (*p* < 0.05) lower than that of the RJ samples (42.19%). A significant (*p* < 0.05) increase in DPPH scavenging capacity was observed in the TJ (45.56~52.77%) samples compared with RJ and PJ. The increase in DPPH scavenging capacity in the juice samples was primarily attributed to an increase in the contents of bioactive compounds (TPC, TFC, and TAC) after thermosonication. In addition, the shear forces generated during ultrasound contributed to the inactivation of some enzymes related to oxidation, which also led to an increase in the antioxidant capacity of the juices [55]. The results of the study on DPPH scavenging capacity showed that TS increased the antioxidant activity of blackcurrant juices and was more effective than pasteurization. This result is consistent with the study of ultrasound on spinach juice [56].

#### 3.3.2. Hydroxyl Radical (·OH) Scavenging Capacity

Hydroxyl radicals are considered to be an important reactive oxygen species that can be formed in biological reactions and lead to cellular senescence and tissue damage through the Fenton reaction [57]. As shown in Figure 3f, the (·OH) scavenging capacity in RJ and PJ was 42.28% and 38.59%, respectively, and the (·OH) scavenging capacity was higher in the TS treatment (44.22~50.85%) than in the PJ treatment. The scavenging capacity of the juice for hydroxyl radicals increased significantly (*p* < 0.05) with increasing TS time, and the temperature change had no significant (*p* > 0.05) effect on the (·OH) in the juice. The enhancement of (·OH) scavenging capacity in the juice may be attributed to the increase in phenolics and anthocyanidin substances during ultrasound treatment [58]. Moreover, the solution experienced an influx of more antioxidants, and the free radical scavenging capacity was enhanced as the ultrasound time increased. Therefore, our results suggest that TS treatment is superior to pasteurization in enhancing the (·OH) scavenging capacity of fruit juices. Previously, similar results were reported in studies on the thermosonication of blueberry juice [59].

In conclusion, the increase in the antioxidant capacity of the TJ samples was closely associated with the increase in the contents of bioactive compounds. It is possible that the generation of shear forces and shock waves capable of destroying the cell wall during TS treatment promotes the release of bioactive compounds from the cells, resulting in an increase in the contents of substances related to antioxidant activity and, thus, an increase in antioxidant capacity [60].

### 3.4. Microbiological Analysis

The results of the juices’ microbiological tests are shown in Table 3. The total bacterial count, along with the yeast and mold counts, reached undetectable levels after pasteurization compared to RJ (4.48 ± 0.01, 4.38 ± 0.01). The inactivation of microorganisms in PJ samples may relate to heat treatment, as high temperatures cause the inactivation of metabolic enzymes and denaturation of proteins, which disrupt the intact structure of the microorganism and change the inherent morphology of certain cells, resulting in irreversible damage to the cells and, ultimately, killing the microorganism [6]. Meanwhile, 40 °C TS treatment significantly (*p* < 0.05) reduced the number of bacteria (3.93 ± 0.03~1.21 ± 0.09 log CFU/mL) and the numbers of yeast and mold (3.59 ± 0.01~1.31 ± 0.03 log CFU/mL), respectively. However, TS reduced the bacterial populations to undetectable levels at 50 and 60 °C, while at 50 °C, TS reduced the mold and yeast counts from 2.31 ± 0.05 to 1.25 ± 0.07 log CFU/mL, and no growth of yeast or mold was observed at 60 °C. These findings were attributed to the localized heating due to cavitation, leading to an increase in system temperature and the formation of hydroxyl radicals, which ultimately led to the inactivation of microorganisms [21]. Similar results were found for the changes in microbial counts in a previous TS study of guava juices [61]. However, incomplete inactivation of yeasts and molds at 40 °C, and partially at 50 °C, may relate to spore formation, since spores can resist this environment. Complete inactivation of microorganisms is achievable with increasing treatment conditions [62].

### 3.5. Sensory Analysis

The sensory evaluation scores ultimately reflect the consumer acceptance of the juices. Table 4 shows that the sensory parameters in the TJ samples were not significantly (*p* > 0.05) different from those in the RJ and PJ samples. The color, taste, flavor, mouthfeel, and overall acceptability in RJ were 6.00 ± 0.20, 5.08 ± 0.08, 4.90 ± 0.10, 5.13 ± 0.06, and 5.73 ± 0.16, respectively, while the PJ samples showed values of 5.95 ± 0.05, 5.03 ± 0.06, 4.90 ± 0.18, 5.00 ± 0.13, and 5.70 ± 0.15, respectively. Compared with RJ and PJ, color variations (5.93 ± 0.06~6.10 ± 0.05), taste (5.07 ± 0.06~5.15 ± 0.05), flavor (4.95 ± 0.05~5.13 ± 0.06), mouthfeel (5.02 ± 0.12~5.13 ± 0.03), and overall acceptability (5.73 ± 0.12~6.03 ± 0.16) were found in the TJ samples. It was observed that the PJ samples had slightly lower sensory scores than all RJ and TJ samples in this study. The reasons for the reduced sensory evaluation scores may be due to the degradation of flavor compounds in the juice caused by pasteurization, or the production of undesirable flavors, color changes due to non-enzymatic browning, and changes in taste due to the deterioration of certain compounds, which can affect the unique taste of blackcurrant juices [63,64]. On the other hand, the samples treated with TS at 40 °C and 50 °C had slightly higher sensory scores compared to the RJ samples; this improvement in sensory properties may be related to the cavitation induced by TS, possibly associated with a reduction in the oxygen content of the juices [65]. In contrast, the sensory scores of the TS treatment at 60 °C were reduced, probably due to the prolonged exposure of the juice to high temperatures during TS and the deterioration of some fresh flavor compounds, which adversely affected the sensory properties of the juices [66]. This is consistent with the results observed in the TS of guava juices [61]. It also demonstrates that the TS preserved the original sensory parameters of the blackcurrant juice without negative impacts.

## 4. Conclusions

This study explored the impacts of thermosonication on the quality attributes of blackcurrant juice, revealing no significant effects on pH, TSS, or TA. However, TS improved the color value, cloudiness, BI, and viscosity of the juice, and it significantly enhanced the contents of bioactive compounds (total phenolic, flavonoid, and anthocyanin contents) with increasing treatment time, while the scavenging capacities for DPPH and (·OH) radicals showed the same trend. The reduction in the AAC of the TS-treated juices was smaller compared with the pasteurization treatment. Significant microbial inactivation was observed in both PJ and TJ samples, with the sensory parameters showing no significant differences. In conclusion, TS is superior to pasteurization in improving the nutritional and sensory qualities of blackcurrant juice, suggesting that it is an effective alternative to pasteurization. Furthermore, the optimal temperature and duration of TS are crucial for achieving high-quality juice. However, the effect of TS on the quality of blackcurrant juice during the storage period needs to be further investigated.

## Figures and Tables

**Figure 1 foods-13-00809-f001:**
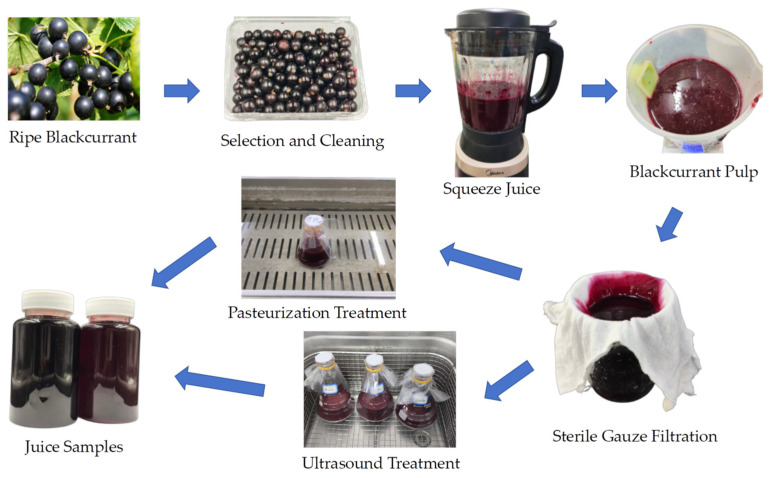
Freshly squeezed blackcurrant juice and treatment processes.

**Figure 2 foods-13-00809-f002:**
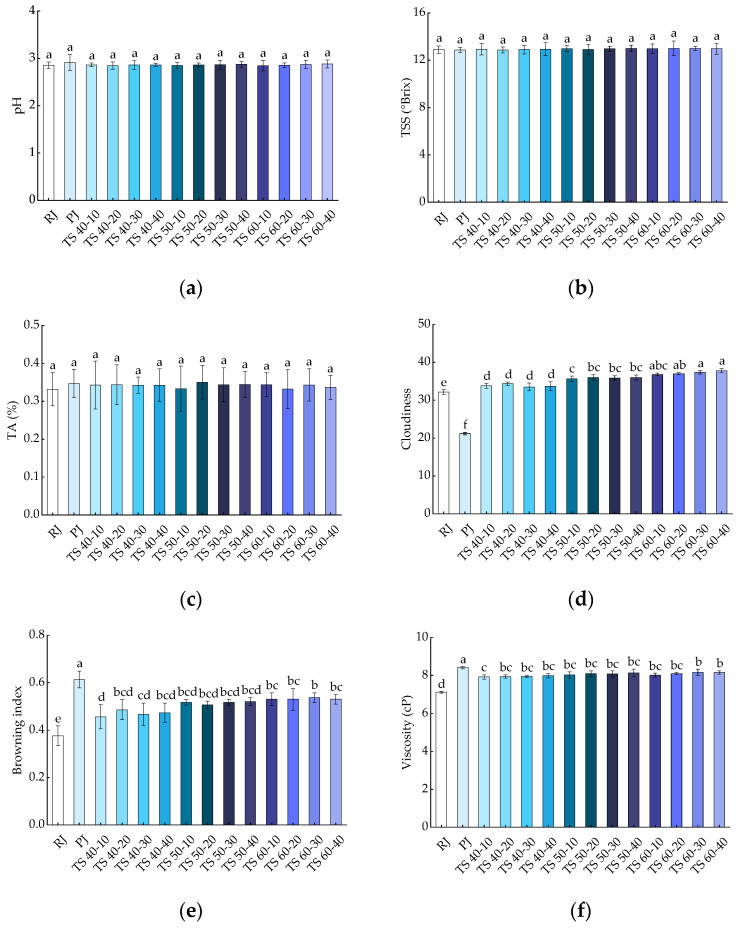
Effects of different treatments on the physicochemical properties of blackcurrant juice: (**a**) pH; (**b**) TSS; (**c**) TA; (**d**) cloudiness; (**e**) browning index; (**f**) viscosity. Different letters indicate significant (*p* < 0.05) differences between treatments.

**Figure 3 foods-13-00809-f003:**
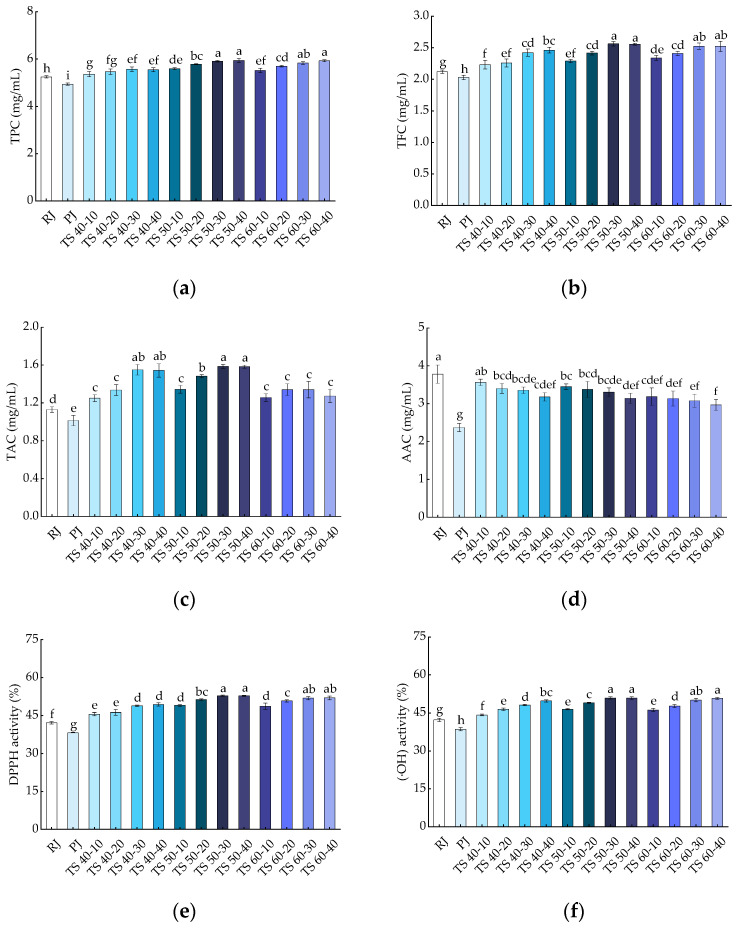
Effects of treatments on bioactive compounds in blackcurrant juice: (**a**) TPC; (**b**) TFC; (**c**) TAC; (**d**) AAC; (**e**) DPPH; (**f**) (·OH). Different letters indicate significant (*p* < 0.05) differences between treatments.

**Table 1 foods-13-00809-t001:** Different parameters for experimental treatments of blackcurrant juice.

Volume	Treatment	Temperature	Time	Power	Frequency
200 mL juice	RJ	-	-	-	-
	PJ	90 °C	1 min	-	-
	TS 40-10	40 °C	10 min	480 W	40 kHz
	TS 40-20		20 min		
	TS 40-30		30 min		
	TS 40-40		40 min		
	TS 50-10	50 °C	10 min		
	TS 50-20		20 min		
	TS 50-30		30 min		
	TS 50-40		40 min		
	TS 60-10	60 °C	10 min		
	TS 60-20		20 min		
	TS 60-30		30 min		
	TS 60-40		40 min		

**Table 2 foods-13-00809-t002:** Changes in the color parameters of blackcurrant juice; RGB represents red, green, and blue; HEX stands for the corresponding color code. Different letters indicate significant (*p* < 0.05) differences between treatments.

Treatment	Appearance	L*	a*	b*	ΔE	RGB	HEX
RJ	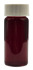	23.81 ± 0.23 ^d^	1.58 ± 0.07 ^a^	−2.46 ± 0.12 ^cd^	-	57, 56, 60	# 39383C
PJ	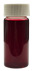	25.83 ± 0.29 ^a^	1.31 ± 0.04 ^cde^	−2.11 ± 0.09 ^a^	2.07 ± 0.06 ^ab^	62, 61, 64	# 3E3D40
TS 40-10	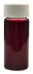	25.06 ± 0.22 ^c^	1.45 ± 0.07 ^b^	−2.41 ± 0.10 ^cd^	1.26 ± 0.23 ^d^	60, 59, 63	# 3C3B3F
TS 40-20	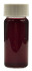	25.24 ± 0.33 ^bc^	1.42 ± 0.13 ^bc^	−2.38 ± 0.09 ^bcd^	1.45 ± 0.51 ^cd^	60, 59, 63	# 3C3B3F
TS 40-30	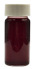	25.61 ± 0.49 ^ab^	1.37 ± 0.04 ^bcd^	−2.31 ± 0.05 ^bc^	1.82 ± 0.27 ^abc^	60, 59, 63	# 3C3B3F
TS 40-40	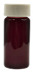	25.70 ± 0.35 ^ab^	1.10 ± 0.09 ^fg^	−2.37 ± 0.13 ^bcd^	2.04 ± 0.12 ^ab^	61, 60, 64	# 3D3C40
TS 50-10	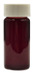	25.31 ± 0.90 ^abc^	1.28 ± 0.14 ^de^	−2.42 ± 0.09 ^cd^	1.95 ± 0.90 ^ab^	60, 60, 64	# 3D3C40
TS 50-20	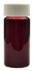	25.79 ± 0.33 ^ab^	1.24 ± 0.02 ^e^	−2.45 ± 0.11 ^cd^	2.01 ± 0.11 ^ab^	61, 61, 65	# 3D3D41
TS 50-30	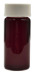	25.67 ± 0.12 ^ab^	1.32 ± 0.05 ^cde^	−2.51 ± 0.05 ^d^	1.89 ± 0.30 ^abc^	61, 60, 65	# 3D3C41
TS 50-40	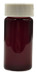	25.74 ± 0.39 ^ab^	1.20 ± 0.04 ^ef^	−2.41 ± 0.11 ^cd^	1.96 ± 0.45 ^ab^	61, 61, 65	# 3D3D41
TS 60-10	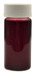	25.73 ± 0.30 ^ab^	1.03 ± 0.06 ^gh^	−2.37 ± 0.06 ^bcd^	1.61 ± 0.29 ^bcd^	61, 61, 65	# 3D3D41
TS 60-20	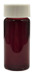	25.76 ± 0.28 ^ab^	0.92 ± 0.04 ^hi^	−2.36 ± 0.03 ^bcd^	2.06 ± 0.09 ^ab^	61, 61, 65	# 3D3D41
TS 60-30	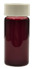	25.71 ± 0.11 ^ab^	0.82 ± 0.04 ^i^	−2.35 ± 0.11 ^bcd^	2.06 ± 0.16 ^ab^	61, 61, 64	# 3D3D40
TS 60-40	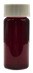	25.77 ± 0.22 ^ab^	0.83 ± 0.03 ^i^	−2.21 ± 0.09 ^ab^	2.12 ± 0.17 ^a^	61, 61, 64	# 3D3D40

**Table 3 foods-13-00809-t003:** Effects of treatments on the numbers of microorganisms in blackcurrant juice; different letters indicate significant (*p* < 0.05) differences between treatments.

Treatment	Total Microbial Count (log CFU/mL)	Yeast and Mold Count (log CFU/mL)
RJ	4.48 ± 0.01 ^a^	4.38 ± 0.01 ^a^
PJ	ND	ND
TS 40-10	3.93 ± 0.03 ^b^	3.59 ± 0.01 ^b^
TS 40-20	3.43 ± 0.05 ^c^	3.22 ± 0.10 ^c^
TS 40-30	2.32 ± 0.07 ^d^	1.85 ± 0.05 ^e^
TS 40-40	1.21 ± 0.09 ^e^	1.31 ± 0.03 ^f^
TS 50-10	ND	2.31 ± 0.05 ^d^
TS 50-20	ND	1.25 ± 0.07 ^f^
TS 50-30	ND	ND
TS 50-40	ND	ND
TS 60-10	ND	ND
TS 60-20	ND	ND
TS 60-30	ND	ND
TS 60-40	ND	ND

**Table 4 foods-13-00809-t004:** Effects of different treatments on sensory parameters of blackcurrant juice; different letters indicate significant (*p* < 0.05) differences between treatments.

Treatment	Color	Taste	Flavor	Mouthfeel	Overall Acceptability
RJ	6.00 ± 0.20 ^a^	5.08 ± 0.08 ^ab^	4.90 ± 0.10 ^b^	5.13 ± 0.06 ^a^	5.73 ± 0.16 ^ab^
PJ	5.95 ± 0.05 ^a^	5.03 ± 0.06 ^b^	4.90 ± 0.18 ^b^	5.00 ± 0.13 ^a^	5.70 ± 0.15 ^b^
TS 40-10	6.10 ± 0.05 ^a^	5.07 ± 0.06 ^ab^	5.13 ± 0.06 ^a^	5.03 ± 0.06 ^a^	5.82 ± 0.20 ^ab^
TS 40-20	5.93 ± 0.06 ^a^	5.08 ± 0.08 ^ab^	5.07 ± 0.06 ^ab^	5.08 ± 0.12 ^a^	5.80 ± 0.10 ^ab^
TS 40-30	6.02 ± 0.03 ^a^	5.11 ± 0.03 ^ab^	4.97 ± 0.06 ^ab^	5.07 ± 0.06 ^a^	5.90 ± 0.20 ^ab^
TS 40-40	5.97 ± 0.08 ^a^	5.11 ± 0.03 ^ab^	5.03 ± 0.12 ^ab^	5.10 ± 0.30 ^a^	5.85 ± 0.05 ^ab^
TS 50-10	6.02 ± 0.08 ^a^	5.11 ± 0.03 ^ab^	4.95 ± 0.05 ^ab^	5.13 ± 0.03 ^a^	5.75 ± 0.15 ^ab^
TS 50-20	6.07 ± 0.08 ^a^	5.12 ± 0.03 ^ab^	5.08 ± 0.08 ^ab^	5.13 ± 0.03 ^a^	6.03 ± 0.16 ^a^
TS 50-30	6.00 ± 0.09 ^a^	5.15 ± 0.05 ^a^	5.07 ± 0.08 ^ab^	5.07 ± 0.03 ^a^	5.92 ± 0.20 ^ab^
TS 50-40	6.08 ± 0.03 ^a^	5.11 ± 0.03 ^ab^	5.07 ± 0.12 ^ab^	5.07 ± 0.03 ^a^	5.82 ± 0.10 ^ab^
TS 60-10	5.95 ± 0.10 ^a^	5.07 ± 0.06 ^ab^	5.02 ± 0.13 ^ab^	5.08 ± 0.06 ^a^	5.80 ± 0.20 ^ab^
TS 60-20	6.03 ± 0.10 ^a^	5.11 ± 0.03 ^ab^	4.98 ± 0.10 ^ab^	5.02 ± 0.12 ^a^	5.88 ± 0.08 ^ab^
TS 60-30	6.05 ± 0.05 ^a^	5.07 ± 0.06 ^ab^	5.02 ± 0.12 ^ab^	5.03 ± 0.12 ^a^	5.87 ± 0.23 ^ab^
TS 60-40	5.93 ± 0.10 ^a^	5.11 ± 0.03 ^ab^	4.97 ± 0.06 ^ab^	5.07 ± 0.08 ^a^	5.73 ± 0.12 ^ab^

## Data Availability

The original contributions presented in this study are included in the article; further inquiries can be directed to the corresponding authors.

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
