# Peer review of "Effects of Thermosonication on the Antioxidant Capacity and Physicochemical, Bioactive, Microbiological, and Sensory Qualities of Blackcurrant Juice"

_foods, 2024, doi:10.3390/foods13050809_

Round 1

Reviewer 1 Report1 Comments and Suggestions for Authors

 General comments

This manuscript evaluates the effect of thermosonication on physicochemical, bioactive, microbiological, and sensory qualities of blackcurrant juice. The study is of interest to the field of juice preservation. The experimental work is in general performed well. However, the effect of the proposed during juice storage is missing. Furthermore, the microbiological evaluation of yeast and bacterial population is not enough to assess the safety of the juice. The authors must revise the legislation and consider the need of  de 5-log pathogen reduction required for alternatives to thermal process.  

 Specific comments

For all the manuscript put reference numbers between parentheses

Define each acronym the first time is it mentioned.

 Materials and Methods

 2.8 Microbiological Analysis: only yeast evaluation is mentioned.

3.4 Color Parameters: add the meaning of changes in parameters and b.

 Results and discussion

 3.5.1 Total Phenolic Content: revise the significant differences between values: 5.25 +/-0.05 is different from 5.36+/-0.11?

 The sensory parameters did not show significant differences among them: However, the discussion is not in accordance with mentioned trend.

Comments on the Quality of English Language

no specific comments

Author Response

Dear reviewer,
As per your kind guidance. We have carefully revised our manuscript according to these suggestions. ‘Point-by-point response letter for reviewer’ have been uploaded to the attachment in word file. Please see the attachment.
We hope the revised manuscript now meets the standard for publishing in Foods.
Thanks for your consideration.
Xiaokun Qiu

Reviewer 2 Report

Comments and Suggestions for Authors

Dear authors, the revised manuscript is interesting, however, it is necessary to address the following recommendations:

Line 2: When it indicates microbial, is it part of the bioactive?

Line 12: When you mention the word bioactive, are you referring to antioxidant activity or also antimicrobial activity? Or refers to bioactive compounds like in line 81

Line 15: the meaning of the abbreviations TSS and TA were no indicated

Line 16: insert spaces in probability value

Line 19: there is or there was

Line 21: It is not necessary to use the abbreviation of the antioxidant methods if they will not be used later in the text of the abstract

Line 23: rewrite… both Pj and Ts?

Line 27: rewrite… properties; bioactive…

Line 38: did you mean [3]

Line 40: did you mean [4,5]

Line 43: did you mean [6]

Line 47: did you mean [7,8]

Line 49: did you mean [9]… [10]

Line 50: did you mean [7]

NOTE: It is necessary to correct the numbering format of the references throughout the document, these must appear between square brackets

Line 77: rewrite… activity [14,24].

Line 81: remove… (pH, TSS, TA, cloudiness, browning index, viscosity, color)

Line 81:

Line 82: remove… (TPC, TFC, TAC, AAC)… (DPPH, ·OH)

Line 85: insert a dot… 2.1. Blackcurrant

Line 103: 4 °C

Line 107: rewrite… Figure 1. Freshly…

Line 109: rewrite… Table 1. Different…

Line 110: in table content insert a space… 1 min

Line 110: in table content use… 30 min instead of 30 min,

Line 112: could include physical-chemical tests in a single section

Line 112: It is important to use the same writing sequence throughout the document… 2.2. pH, Total Soluble Solids (TSS), and Titratable Acidity (TA)

Line 113: rewrite… pH, TSS and TA were…

Line 118: mol/L

Line 119: move pH procedure to line 114

Line 122: Do not use names or surnames of authors, only the reference number should be indicated in square brackets.

Line 124: indicate the temperature used during centrifugation process

Line 140: remove italic text format for L*

Line 145: insert a dot… 2.6. Bioactive

Line 146: rewrite… 2.6.1. Total phenolics content (TPC)

Line 148: Folinol or Folin reagent?

Line 153: rewrite… 2.6.2. Total flavonoid content (TFC)

Line 155: rewrite format like in line 148… (0.3 mL, 5 %)

Line 156: rewrite… (0.6 mL, 10 %)

Line 157: rewrite… (1 mL, 1 M)

Line 161: rewrite… 2.6.3. Total anthocyanin content (TAC)

Line 164: mol/L

Line 165: mol/L

Line 175: mg/100 g

Line 176: rewrite… 2.6.4. Ascorbic acid content (AAC)

Line 178: idem

Line 182: mg/100 g

Line 187: mg/mL

Line 189: This property is not indicated in the title of the work

Line 189: rewrite… 2.7. Antioxidant Capacity

Line 190: rewrite… 2.7.1. DPPH radical scavenging ability

Line 191: idem… authors

Line 192: mmol/L

Line 194: 30 min

Line 199: rewrite… 2.7.2. Hydroxyl radical (·OH) scavenging ability

Line 200: idem… authors

Line 201: 2 mL

Line 202: mol/L

Line 202: 2 mol/L

Line 204: 30 min

Line 209: insert a dot… 2.8.

Line 213: 48 h

Line 213: 37 ± 1 °C or 37 °C ± 1 °C like in line 204

Line 216: CFU/mL

Line 217: insert a dot… 2.9.

Line 219: idem… authors

Line 231: Results and Discussion?

Line 232: insert a dot… 3.1.

Line 253: insert a dot… 3.2.

Line 270: insert a dot… 3.3.

Line 280: insert a dot… 3.4.

Line 283: insert space… RJ (23.81

Line 285: insert space… PJ (25.83

Line 285: insert space… TJ (25.06

Line 297: Error! Reference source not found..?

Line 298: insert a dot… Table 2. Changes

Line 298: You could insert a column on the right side of the table and obtain the RGB and HEX code values and insert the visual color

Line 301: insert a dot… 3.5.

Line 306: rewrite… 3.5.1. TPC…. It was previously abbreviated

Line 310: mg/mL

Line 325: in figure content use mg/mL instead of mg*mL-1

Line 328: The meaning of the literals is not indicated, like the previous figure.

Line 329: rewrite… 3.5.2. TFC…. It was previously abbreviated

Line 332: mg/mL

Line 334: mg/mL

Line 351: rewrite… 3.5.3. TAC…. It was previously abbreviated

Line 355: mg/mL

Line 357: mg/mL… modify through the document

Line 369: rewrite… 3.5.4. AAC…. It was previously abbreviated

Line 381: Error! Reference source not found..??

Line 387: insert a dot… 3.6.

Line 388: rewrite… 3.6.1. DPPH radical scavenging ability

Line 401: rewrite… 3.6.2. Hydroxyl radical (·OH) scavenging capacity

Line 422: insert a dot… 3.7.

Line 430,431,433: CFU/mL… modify through the document

Line 441: insert a dot… Table 3. Effect

Line 445: insert a dot… 3.8.

Line 457: Error! Reference source not found..??

Line 468: insert a dot… Table 4. Effect

Line 472: In conclusions it is not necessary to use probability values

Line 503: references section… use the proper format to cite references

Author Response

(The authors gave the same response as above.)

Round 2

Reviewer 2 Report

Comments and Suggestions for Authors

Dear authors, the modifications to the manuscript were attended to, except that in the references section it is required to address the following:

Considering the following reference format described in Microsoft word template:

Author 1, A.B.; Author 2, C.D. Title of the article. Abbreviated Journal Name Year, Volume, page range.

Line 504: rewrite…. Cao, L.; Park, Y.; Lee, S.; Kim, D-O. Extraction

Line 505: text format… 11(4)

Line 505: use Appl. Sci. instead of J. Appl. Sci.

Line 506,508,510,512,514,516,521,523,525,528,530,532,534,536,538: It is necessary to complete the information of the reference authors and use the correct text format as described in Microsoft word template

Line 507,509,536,553: use LWT. Instead of J. LWT. Note: make this modification through this section

Line 509: text format… 49(2)

Line 513: use Food Chem. Instead of J. Food Chem.

Line 515: use 26-37 instead of 26–37

Line 515: use Trends Food Sci. Technol. Instead of J. Trends Food Sci. Technol.

Line 517,539,549: use the correct abbreviation… IFSET.

Line 517: remove space… (Citrus

Line 519,546: It is necessary to use the correct text format for authors names as described in Microsoft word template

Line 520,524,526,535,576: did you mean Food Chem. Note: make this modification through this section

Line 520: 96–100

Line 522,568,570: did you mean Food Biosci.

Line 524: 150–161

Line 526: 422–430

Line 529: insert the correct journal name of this reference

Line 531,542,550,561,566: did you mean Ultrason. Sonochem. Note: make this modification through this section

Line 531: 540–560

Line 533: Appl. Food Res.

Line 533: text format… 2(2)

Line 535,540: include the reference volume

Line 539: J. Innov. Food Sci. Emerg. Technol.

Line 541,544,548: It is necessary to complete the information of the reference authors and use the correct text format as described in Microsoft word template

Line 543: 277–286

Line 545: text format… 50(5)

Line 545: Int. J. Food Sci. Technol.

Line 545: 1275–1282

Line 546: use italic text format for scientific names

Line 547: 39(6)  …. Note: use this format for the following references

Line 547: 1744–1753.

Line 549: 186–195. Note: use this format for the following references

Line 550: It is necessary to complete the information of the reference authors and use the correct text format as described in Microsoft word template…. Note: it is necessary attend this suggestion for the following references

Line 556: Biosyst Eng.

Line 556: (prepublish)…. Complete the information where this note appears in parentheses in this section, the information is available online

Line 559: Molecules.

Line 564: juice. J. Food Res. Int.

Line 574: research. Food Chem: X.

Line 582: juice. Measurement: Food.

Line 584: use italic text format for scientific names

Line 586: juice. Food Bioprod. Process. Note: check the correct abbreviation for each of the names of the journals of each reference

Author Response

(The authors gave the same response as above.)
